# Tree-structured Gaussian Process Approximations

**Thang Bui**
tdb40@cam.ac.uk

**Richard Turner**
ret26@cam.ac.uk

Computational and Biological Learning Lab, Department of Engineering
University of Cambridge, Trumpington Street, Cambridge, CB2 1PZ, UK

## Abstract

Gaussian process regression can be accelerated by constructing a small pseudo-dataset to summarize the observed data. This idea sits at the heart of many approximation schemes, but such an approach requires the number of pseudo-datapoints to be scaled with the range of the input space if the accuracy of the approximation is to be maintained. This presents problems in time-series settings or in spatial datasets where large numbers of pseudo-datapoints are required since computation typically scales quadratically with the pseudo-dataset size. In this paper we devise an approximation whose complexity grows linearly with the number of pseudo-datapoints. This is achieved by imposing a tree or chain structure on the pseudo-datapoints and calibrating the approximation using a Kullback-Leibler (KL) minimization. Inference and learning can then be performed efficiently using the Gaussian belief propagation algorithm. We demonstrate the validity of our approach on a set of challenging regression tasks including missing data imputation for audio and spatial datasets. We trace out the speed-accuracy trade-off for the new method and show that the frontier dominates those obtained from a large number of existing approximation techniques.

## 1   Introduction

Gaussian Processes (GPs) provide a flexible nonparametric prior over functions which can be used as a probabilistic module in both supervised and unsupervised machine learning problems. The applicability of GPs is, however, severely limited by a burdensome computational complexity. For example, this paper will consider non-linear regression on a dataset of size $N$ for which training scales as $\mathcal{O}(N^3)$ and prediction as $\mathcal{O}(N^2)$. This represents a prohibitively large computational cost for many applications. Consequently, a substantial research effort has sought to develop efficient approximation methods that side-step these significant computational demands [1–9]. Many of these approximation methods are based upon an intuitive idea, which is to use a smaller pseudo-dataset of size $M \ll N$ to summarize the observed dataset, reducing the cost for training and prediction (typically to $\mathcal{O}(NM^2)$ and $\mathcal{O}(M^2)$). The methods can be usefully categorized into two non-exclusive classes according to the way in which they arrive at the pseudo-dataset. *Indirect posterior approximations* employ a modified generative model that is carefully constructed to be calibrated to the original, but for which inference is computationally cheaper. In practice this leads to parametric probabilistic models that inherit some of the GP's robustness to over-fitting. *Direct posterior approximations*, on the other hand, cut to the chase and directly calibrate an approximate posterior distribution, chosen to have favourable computational properties, to the true posterior distribution. In other words, the non-parametric model is retained, but the pseudo-datapoints provide a bottleneck at the inference stage, rather than at the modelling stage.

Pseudo-datapoint approximations have enabled GPs to be deployed in a far wider range of problems than was previously possible. However, they have a severe limitation which means many challenging datasets still remain far out of their reach. The problem arises from the fact that pseudo-dataset methods are functionally local in the sense that each pseudo-datapoint sculpts out the approximate

posterior in a small region of the input space around it [10]. Consequently, when the range of the inputs is large compared to the range of the dependencies in the posterior, many pseudo-datapoints are required to maintain the accuracy of the approximation. In time-series settings [11–13], such as audio denoising and missing data imputation considered later in the paper, this means that the number of pseudo-datapoints must grow with the number of datapoints if restoration accuracy is to be maintained. In other words, $M$ must be scaled with $N$ and so pseudo-datapoint schemes have not reduced the scaling of the computational complexity. In this context, approximation methods built from a series of local GPs are perhaps more appropriate, but they suffer from discontinuities at the boundaries that are problematic in many contexts, in the audio restoration example they lead to audible artifacts. The limitations of pseudo-datapoint approximations are not restricted to the time-series setting. Many datasets in geostatistics, climate science, astronomy and other fields have large, and possibly growing, spatial extent compared to the posterior dependency length. This puts them well out of the reach of all current pseudo-datapoint approximation methods.

The purpose of this paper is to develop a new pseudo-datapoint approximation scheme which can be applied to these challenging datasets. Since the need to scale the number of pseudo-datapoints with the range of the inputs appears to be unavoidable, the approach instead focuses on reducing the computational cost of training and inference so that it is truely linear in $N$. This reduction in computational complexity comes from an indirect posterior approximation method which imposes additional structural restrictions on the pseudo-dataset so that it has a chain or tree structure. The paper is organized as follows: In the next section we will briefly review GP regression together with some well known pseudo-datapoint approximation methods. The tree-structured approximation is then proposed, related to previous methods, and developed in section 2. We demonstrate that this new approximation is able to tractably handle far larger datasets whilst maintaining the accuracy of prediction and learning in section 3.

## 1.1 Regression using Gaussian Processes

This section provides a concise introduction to GP regression [14]. Suppose we have a training set comprising $N$ $D$-dimensional input vectors $\{\mathbf{x}_n\}_{n=1}^N$ and corresponding real valued scalar observations $\{y_n\}_{n=1}^N$. The GP regression model assumes that each observation $y_n$ is formed from an unknown function $f(.)$, evaluated at input $\mathbf{x}_n$, which is corrupted by independent Gaussian noise. That is $y_n = f(\mathbf{x}_n) + \epsilon_n$ where $p(\epsilon_n) = \mathcal{N}(\epsilon_n; 0, \sigma^2)$. Typically a zero mean GP is used to specify a prior over the function $f$ so that any finite set of function values are distributed under the prior according to a multivariate Gaussian $p(\mathbf{f}) = \mathcal{N}(\mathbf{f}; \mathbf{0}, \mathbf{K_{ff}})$.[1] The covariance of this Gaussian is specified by a covariance function or kernel, $(\mathbf{K_{ff}})_{n,n'} = k_\theta(\mathbf{x}_n, \mathbf{x}_{n'})$, which depends upon a small number of hyper-parameters $\theta$. The form of the covariance function and the values of the hyper-parameters encapsulates prior knowledge about the unknown function. Having specified the probabilistic model, we now consider regression tasks which typically involve predicting the function value $f_*$ at some unseen input $\mathbf{x}_*$ (also known as missing data imputation) or estimating the function value $f$ at a training input $\mathbf{x}_n$ (also known as denoising). Both of these prediction problems can be handled elegantly in the GP regression framework by noting that the posterior distribution over the function values is another Gaussian process with a mean and covariance function given by

$$m_f(\mathbf{x}) = K_{\mathbf{xf}}(\mathbf{K_{ff}} + \sigma^2 \mathbf{I})^{-1}\mathbf{y}, \quad k_f(\mathbf{x}, \mathbf{x}') = k(\mathbf{x}, \mathbf{x}') - K_{\mathbf{xf}}(\mathbf{K_{ff}} + \sigma^2 \mathbf{I})^{-1}K_{\mathbf{fx}'}. \quad (1)$$

Here $\mathbf{K_{ff}}$ is the covariance matrix on the training set defined above and $K_{\mathbf{xf}}$ is the covariance function evaluated at pairs of test and training inputs. The hyperparameters $\theta$ and the noise variance $\sigma^2$ can be learnt by finding a (local) maximum of the marginal likelihood of the parameters, $p(\mathbf{y}|\theta, \sigma) = \mathcal{N}(\mathbf{y}; \mathbf{0}, \mathbf{K_{ff}} + \sigma^2 \mathbf{I})$. The origin of the cubic computational cost of GP regression is the need to compute the Cholesky decomposition of the matrix $\mathbf{K_{ff}} + \sigma^2 \mathbf{I}$. Once this step has been performed a subsequent prediction can be made in $\mathcal{O}(N^2)$.

## 1.2 Review of Gaussian process approximation methods

There are a plethora of methods for accelerating learning and inference in GP regression. Here we provide a brief and inexhaustive survey that focuses on indirect posterior approximation schemes based on pseudo-datasets. These approximations can be understood in terms of a three stage process. In the first stage the generative model is augmented with pseudo-datapoints, that is a set of pseudo-input points $\{\bar{\mathbf{x}}_m\}_{m=1}^M$ and (noiseless) pseudo-observations $\{u_m\}_{m=1}^M$. In the second stage

some of the dependencies in the model prior distribution are removed so that inference becomes computationally tractable. In the third stage the parameterisation of the new model is chosen in such a way that it is calibrated to the old one. This last stage can seem mysterious, but it can often be usefully understood as a KL divergence minimization between the true and the modified model.

Perhaps the simplest example of this general approach is the Fully Independent Training Conditional (FITC) approximation [4] (see table 1). FITC removes direct dependencies between the function values $\mathbf{f}$ (see fig. 1) and calibrates the modified prior using the KL divergence $\mathrm{KL}(p(\mathbf{f}, \mathbf{u}) \| q(\mathbf{f}, \mathbf{u}))$ yielding $q(\mathbf{f}, \mathbf{u}) = p(\mathbf{u}) \prod_{n=1}^{N} p(f_n|\mathbf{u})$. That this model leads to computational advantages can perhaps most easily be seen by recognising that it is essentially a factor analysis model, with an admittedly clever parameterisation in terms of the covariance function. FITC has since been extended so that the pseudo-datapoints can have a different covariance function to the data [6] and so that some subset of the direct dependencies between the function values $\mathbf{f}$ are retained as in the Partially Independent Conditional (PIC) approximation [3,5] which generalizes the Bayesian Committee Machine [15].

There are indirect approximation methods which do not naturally fall into this general scheme. Stationary covariance functions can be approximated using a sum of $M$ cosines which leads to the Sparse Spectrum Gaussian Process (SSGP) [7] which has identical computational cost to FITC. An alternative prior approximation method for stationary covariance functions in the multi-dimensional time-series setting designs a linear Gaussian state space model (LGSSM) so that it approximates the prior power spectrum using a connection to stochastic differential equations (SDEs) [16]. The Kalman smoother can then be used to perform inference and learning in the new representation with a linear complexity. This technique, however, only reduces the computational complexity for the temporal axis and the spatial complexity is still cubic, moreover the extension beyond the time-series setting requires a second layer of approximations, such as variational free-energy methods [17] which are known to introduce significant biases [18].

In contrast to the methods mentioned above, direct posterior approximation methods do not alter the generative model, but rather seek computational savings through a simplified representation of the posterior distribution. Examples of this type of approach include the Projected Process (PP) method [1, 2] which has been since been interpreted as the expectation step in a variational free energy (VFE) optimisation scheme [8] enabling stochastic versions [19]. Similarly, the Expectation Propagation (EP) framework can also be used to devise posterior approximations with associated hyper-parameter learning scheme [9]. All of these methods employ a pseudo-dataset to parameterize the approximate posterior.

| Method | KL minimization | Result |
|---|---|---|
| FITC* | $\mathrm{KL}(p(\mathbf{f}, \mathbf{u}) \| q(\mathbf{u}) \prod_n q(f_n|\mathbf{u}))$ | $q(\mathbf{u}) = p(\mathbf{u}), q(f_n|\mathbf{u}) = p(f_n|\mathbf{u})$ |
| PIC* | $\mathrm{KL}(p(\mathbf{f}, \mathbf{u}) \| q(\mathbf{u}) \prod_k q(\mathbf{f}_{C_k}|\mathbf{u}))$ | $q(\mathbf{u}) = p(\mathbf{u}), q(\mathbf{f}_{C_k}|\mathbf{u}) = p(\mathbf{f}_{C_k}|\mathbf{u})$ |
| PP | $\mathrm{KL}(\frac{1}{Z}p(\mathbf{u})p(\mathbf{f}|\mathbf{u})q(\mathbf{y}|\mathbf{u}) \| p(\mathbf{f}, \mathbf{u}|\mathbf{y}))$ | $q(\mathbf{y}|\mathbf{u}) = \mathcal{N}(\mathbf{y}; \mathbf{K}_{\mathbf{fu}}\mathbf{K}_{\mathbf{uu}}^{-1}\mathbf{u}, \sigma^2\mathbf{I})$ |
| VFE | $\mathrm{KL}(p(\mathbf{f}|\mathbf{u})q(\mathbf{u}) \| p(\mathbf{f}, \mathbf{u}|\mathbf{y}))$ | $q(\mathbf{u}) \propto p(\mathbf{u}) \exp(\langle \log(p(\mathbf{y}|\mathbf{f}))\rangle_{p(\mathbf{f}|\mathbf{u})})$ |
| EP | $\mathrm{KL}(q(\mathbf{f}; \mathbf{u})p(y_n|f_n)/q_n(\mathbf{f}; \mathbf{u}) \| q(\mathbf{f}; \mathbf{u}))$ | $q(\mathbf{f}; \mathbf{u}) \propto p(\mathbf{f}) \prod_m p(u_m|f_m)$ |
| Tree* | $\mathrm{KL}(p(\mathbf{f}, \mathbf{u}) \| \prod_k q(\mathbf{f}_{C_k}|\mathbf{u}_{B_k}) \times$ | $q(\mathbf{f}_{C_k}|\mathbf{u}_{B_k}) = p(\mathbf{f}_{C_k}|\mathbf{u}_{B_k})$ |
| | $q(\mathbf{u}_{B_k}|\mathbf{u}_{\mathrm{par}(B_k)}))$ | $q(\mathbf{u}_{B_k}|\mathbf{u}_{\mathrm{par}(B_k)}) = p(\mathbf{u}_{B_k}|\mathbf{u}_{\mathrm{par}(B_k)})$ |

Table 1: GP approximations as KL minimization. $C_k$ and $B_k$ are disjoint subsets of the function values and pseudo-datapoints respectively. Indirect posterior approximations are indicated $*$.

## 1.3 Limitations of current pseudo-dataset approximations

There is a conflict at the heart of current pseudo-dataset approximations. Whilst the effect of each pseudo-datapoint is *local*, the computations involving them are *global*. The local characteristic means that large numbers of pseudo-datapoints are required to accurately approximate complex posterior distributions. If $l_d$ is the range of the dependencies in the posterior in dimension $d$ and $L_d$ is the data-range in each dimension then approximation accuracy will be retained when $M \gtrsim \prod_{d=1}^{D} L_d/l_d$. Critically, for many applications this condition means that large numbers of pseudo-points are required, such as time series ($L_1 \propto N$) and large spatial datasets ($L_d \gg l_d$). Unfortunately, the global graphical structure means that it is computationally costly to handle such large pseudo-datasets. The obvious solution to this conflict is to use the so-called local approximation which splits the observations into disjoint blocks and models each one with a GP. This is a severe approach and this paper

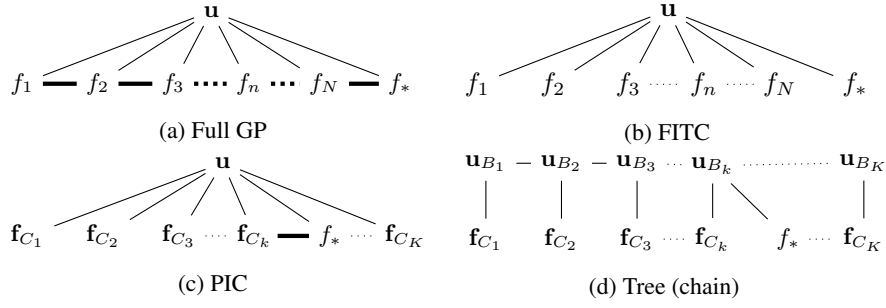

(a) Full GP      (b) FITC

(c) PIC      (d) Tree (chain)

Figure 1: Graphical models of the GP model and different prior approximation schemes using pseudo-datapoints. Thick edges indicate full pairwise connections and boldface fonts denote sets of variables. The chain structured version of the new approximation is shown for clarity.

proposes a more elegant and accurate alternative that retains more of the graphical structure whilst still enabling local computation.

## 2   Tree-structured prior approximations

In this section we develop an indirect posterior approximation in the same family as FITC and PIC. In order to reduce the computational overhead of these approximations, the global graphical structure is replaced by a local one via two modifications. First, the $M$ pseudo-datapoints are divided into $K$ disjoint blocks of potentially different cardinality $\{\mathbf{u}_{B_k}\}_{k=1}^K$ and the blocks are then arranged into a tree. Second, the function values are also divided into $K$ disjoint blocks of potentially different cardinality $\{\mathbf{f}_{C_k}\}_{k=1}^K$ and the blocks are assumed to be conditionally independent given the corresponding subset of pseudo-datapoints. The new graphical model is shown in fig. 1d and it can be described mathematically as follows,

$$q(\mathbf{u}) = \prod_{k=1}^K q(\mathbf{u}_{B_k}|\mathbf{u}_{\text{par}(B_k)}), \quad q(\mathbf{f}|\mathbf{u}) = \prod_{k=1}^K q(\mathbf{f}_{C_k}|\mathbf{u}_{B_k}), \quad p(\mathbf{y}|\mathbf{f}) = \prod_{n=1}^N p(y_n; f_n, \sigma^2). \quad (2)$$

Here $\mathbf{u}_{\text{par}(B_k)}$ denotes the pseudo-datapoints in the parent node of $\mathbf{u}_{B_k}$. This is an example of prior approximation as the original likelihood function has been retained.

The next step is to calibrate the new approximate model by choosing suitable values for the distributions $\{q(\mathbf{u}_{B_k}|\mathbf{u}_{\text{par}(B_k)}), q(\mathbf{f}_{C_k}|\mathbf{u}_{B_k})\}_{k=1}^K$. Taking an identical approach to that employed by FITC and PIC, we minimize a forward KL divergence between the true model prior and the approximation, $\text{KL}(p(\mathbf{f}, \mathbf{u})||\prod_k q(\mathbf{f}_{C_k}|\mathbf{u}_{B_k})q(\mathbf{u}_{B_k}|\mathbf{u}_{\text{par}(B_k)}))$ (see table 1). The optimal distributions are found to be the corresponding conditional distributions in the unapproximated augmented model,

$$q(\mathbf{u}_{B_k}|\mathbf{u}_{\text{par}(B_k)}) = p(\mathbf{u}_{B_k}|\mathbf{u}_{\text{par}(B_k)}) = \mathcal{N}(\mathbf{u}_{B_k}; \mathbf{A}_k \mathbf{u}_{\text{par}(B_k)}, \mathbf{Q}_k), \quad (3)$$

$$q(\mathbf{f}_{C_k}|\mathbf{u}_{B_k}) = p(\mathbf{f}_{C_k}|\mathbf{u}_{B_k}) = \mathcal{N}(\mathbf{f}_{C_k}; \mathbf{C}_k \mathbf{u}_{B_k}, \mathbf{R}_k). \quad (4)$$

The parameters depend upon the covariance function. Letting $\mathbf{u}_k = \mathbf{u}_{B_k}$, $\mathbf{u}_l = \mathbf{u}_{\text{par}(B_k)}$ and $\mathbf{f}_k = \mathbf{f}_{C_k}$ we find that,

$$\mathbf{A}_k = \mathbf{K}_{\mathbf{u}_k \mathbf{u}_l} \mathbf{K}_{\mathbf{u}_l \mathbf{u}_l}^{-1}, \quad \mathbf{Q}_k = \mathbf{K}_{\mathbf{u}_k \mathbf{u}_k} - \mathbf{K}_{\mathbf{u}_k \mathbf{u}_l} \mathbf{K}_{\mathbf{u}_l \mathbf{u}_l}^{-1} \mathbf{K}_{\mathbf{u}_l \mathbf{u}_k}, \quad (5)$$

$$\mathbf{C}_k = \mathbf{K}_{\mathbf{f}_k \mathbf{u}_k} \mathbf{K}_{\mathbf{u}_k \mathbf{u}_k}^{-1}, \quad \mathbf{R}_k = \mathbf{K}_{\mathbf{f}_k \mathbf{f}_k} - \mathbf{K}_{\mathbf{f}_k \mathbf{u}_k} \mathbf{K}_{\mathbf{u}_k \mathbf{u}_k}^{-1} \mathbf{K}_{\mathbf{u}_k \mathbf{f}_k}. \quad (6)$$

As shown in the graphical model, the local pseudo-data separate test and training latent functions. The marginal posterior distribution of the local pseudo-data is then sufficient to obtain the approximate predictive distribution: $p(f_*|\mathbf{y}) = \int d\mathbf{u}_{B_k} p(f_*, \mathbf{u}_{B_k}|\mathbf{y}) = \int d\mathbf{u}_{B_k} p(f_*|\mathbf{u}_{B_k}) p(\mathbf{u}_{B_k}|\mathbf{y})$. In other words, once inference has been performed, prediction is local and therefore fast. The important question of how to assign test and training points to blocks is discussed in the next section.

We note that the tree-based prior approximation includes as special cases; the full GP, PIC, FITC, the local method and local versions of PIC and FITC (see table 1 in the supplementary material). Importantly, in a time-series setting the blocks can be organized into a chain and the approximate model becomes a LGSSM. This provides an new method for approximating GPs using LGSSMs in which the state is a set pseudo-observations, rather than for instance, the derivatives of function values at the input locations [16].

Exact inference in this approximate model proceeds efficiently using the up-down algorithm for Gaussian Beliefs (see [20, Ch. 14]). The inference scheme has the same complexity as forming the model, $\mathcal{O}(KD^3) \approx \mathcal{O}(ND^2)$ (where $D$ is the average number of observations per block).

## 2.1 Inference and learning

**Selecting the pseudo-inputs and constructing the tree** First we consider the method for dividing the observed data into blocks and selecting the pseudo-inputs. Typically, the block sizes will be chosen to be fairly small in order to accelerate learning and inference. For data which are on a grid, such as regularly sampled time-series considered later in the paper, it may be simplest to use regular blocks. An alternative, which might be more appropriate for non-regularly sampled data, is to use a k-means algorithm with the Euclidean distance score. Having blocked the observations, a random subset of the data in each block are chosen to set the pseudo-inputs. Whilst it would be possible in principle to optimize the locations of the pseudo-inputs, in practice the new approach can tractably handle a very large number of pseudo-datapoints (e.g. $M \approx N$), and so optimisation is less critical than for previous approaches. Once the blocks are formed, they are fixed during hyperparameter training and prediction. Second, we consider how to construct the tree. The pair-wise distances between the cluster centers are used to define the weights between candidate edges in a graph. Kruskal's algorithm uses this information to construct an acyclic graph. The algorithm starts with a fully disconnected graph and recursively adds the edge with the smallest weight that does not introduce loops. A tree is randomly formed from this acyclic subgraph by choosing one node to be the root. This choice is arbitrary and does not affect the results of inference. The parameters of the model $\{\mathbf{A}_k, \mathbf{Q}_k, \mathbf{C}_k, \mathbf{R}_k\}_{k=1}^K$ (state transitions and noise) are computed by traversing down the tree from the root to the leaves. These matrices must be recomputed at each step during learning.

**Inference** It is straightforward to marginalize out the latent functions $\mathbf{f}$ in the graphical model in which case the effective local likelihood becomes $p(\mathbf{y}_k|\mathbf{u}_k) = \mathcal{N}(\mathbf{y}_k; \mathbf{C}_k\mathbf{u}_k, \mathbf{R}_k + \sigma^2\mathbf{I})$. The model can be recognized from the graphical model as a tree-structured Gaussian model with latent variables $\mathbf{u}$ and observations $\mathbf{y}$. As is shown in the supplementary, the posterior distribution can be found by using the Gaussian belief propagation algorithm (for more see [20]). The passing of messages can be scheduled so the marginals can be found after two passes (asynchronous scheduling: upwards from leaves to root and then downwards). For chain structures inference can be performed using the Kalman smoother at the same cost.

**Hyperparameter learning** The marginal likelihood can be efficiently computed by the same belief propagation algorithms due to its recursive form, $p(\mathbf{y}_{1:K}|\theta) = \prod_{k=1}^K p(\mathbf{y}_k|\mathbf{y}_{1:k-1}, \theta)$. The derivatives can also be tractably computed as they involve only local moments:

$$\frac{\mathrm{d}}{\mathrm{d}\theta} \log p(\mathbf{y}|\theta) = \sum_{k=1}^K \left[ \langle \frac{\mathrm{d}}{\mathrm{d}\theta} \log p(\mathbf{u}_k|\mathbf{u}_l) \rangle_{p(\mathbf{u}_k, \mathbf{u}_l|\mathbf{y})} + \langle \frac{\mathrm{d}}{\mathrm{d}\theta} \log p(\mathbf{y}_k|\mathbf{u}_k) \rangle_{p(\mathbf{u}_k|\mathbf{y})} \right]. \qquad (7)$$

For concreteness, the explicit form of the marginal likelihood and its derivative are included in the supplementary material. We obtain point estimates of the hyperparameters by finding a (local) maximum of the marginal likelihood using the BFGS algorithm.

## 3 Experiments

We test the new approximation method on three challenging real-world prediction tasks[2] via a speed-accuracy trade-off as recommended in [21]. Following that work, we did not investigate the effects of pseudo-input optimisation. We used different datasets that had less limited spatial/temporal extent.

**Experiment 1: Audio sub-band data (exponentiated quadratic kernel)** In the first experiment we consider imputation of missing data in a sub-band of a speech signal. The speech signal was taken from the TIMIT database (see fig. 4), a short time Fourier transform was applied (20ms Gaussian window), and the real part of the 152Hz channel selected for the experiments. The signal was $T = 50000$ samples long and 25 sections of length 80 samples were removed. An exponentiated quadratic kernel, $k_\theta(t, t') = \sigma^2 \exp(-\frac{1}{2l^2}(t - t')^2)$, was used for prediction. We compare the chain

structured pseudo-datapoint approximation to FITC, VFE, SSGP, local versions of PIC (corresponding to setting $\mathbf{A}_k = \mathbf{0}$, $\mathbf{Q}_k = \mathbf{K}_{\mathbf{u}_k \mathbf{u}_k}$ in the tree-structured approximation) and the SDE method.[3] Only 20000 datapoints were used for the SDE method due to the long run times. The size of the pseudo-dataset and the number of blocks in the chain and local approximations, and the order of approximation in SDE were varied to trace out speed-accuracy frontiers. Accuracy of the imputation was quantified using the standardized mean squared errors (SMSEs) (for other metrics, see the supplementary material). Hyperparameter learning proceeded until a convergence criteria or a maximum number of function evaluations was reached. Learning and prediction (imputation) times were recorded. We found that the chain structured method outperforms all of the other methods (see fig. 2). For example, for a fixed training time of 100s, the best performing chain provided a three-fold increase in accuracy over the local method which was the next best. A typical imputation is shown in fig. 4 (left hand side). The chain structured method was able to accurately impute the missing data whilst that the local method is less accurate and more uncertain as information is not propagated between the blocks.

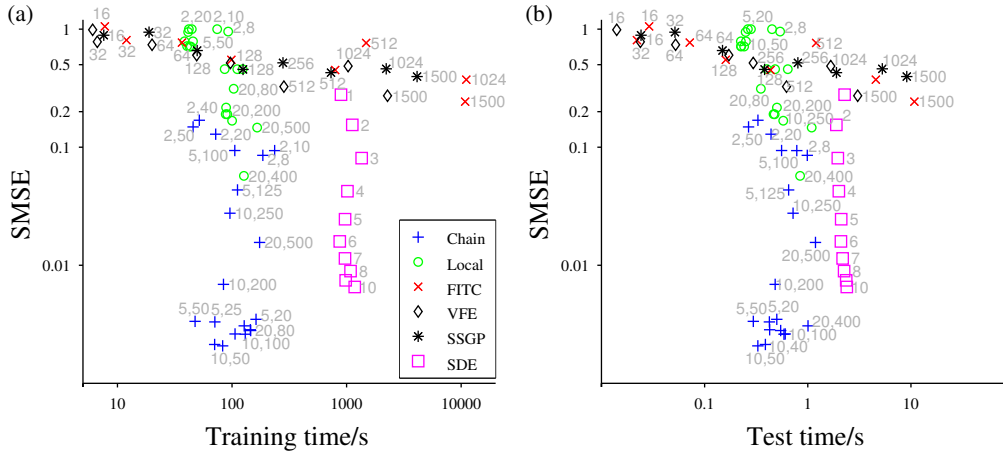

Figure 2: Experiment 1. Audio sub-band reconstruction error as a function of training time (a) and test time (b) for different approximations. The numerical labels for the *chain* and *local* methods are the number of pseudo-datapoints per block and the number of observations per block respectively, and for the SDE method are the order of approximation. For the other methods they are the size of the pseudo-dataset. Faster and more accurate approximations are located towards the bottom left hand corners of the plots.

**Experiment 2: Audio filter data (spectral mixture)**   The second experiment tested the performance of the chain based approximation when more complex kernels are employed. We filtered the same speech signal using a 152Hz filter with a 50Hz bandwidth, producing a signal of length $T = 50000$ samples from which missing sections of length 150 samples were removed. Since the complete signal had a complex bandpass spectrum we used a spectral mixture kernel containing two components [22], $k_\theta(t, t') = \sum_{k=1}^{2} \sigma_k^2 \cos(\omega_k(t - t')) \exp(-\frac{1}{2l_k^2}(t - t')^2)$. We compared a chain based approximation to FITC, VFE and the local PIC method finding it to be substantially more accurate than both methods (see fig. 3 for SMSE results and the right hand side of fig. 4 for a typical example). Results with more components showed identical trends (see supplementary material).

**Experiment 3: Terrain data (two dimensional input space, exponentiated quadratic kernel)** In the final experiment we tested the tree based appoximation using a spatial dataset in which terrain altitude was measured as a function of geographical position.[4] We considered a 20km by 30km region (400×600 datapoints) and tested prediction on 80 randomly positioned missing blocks of size 1km by 1km (20x20 datapoints). In total, this translates into about 200k/40k training/test points. We used an exponentiated quadratic kernel with different length-scales in the two input dimensions, comparing a tree-based approximation, which was constructed as described in section 2.1, to the

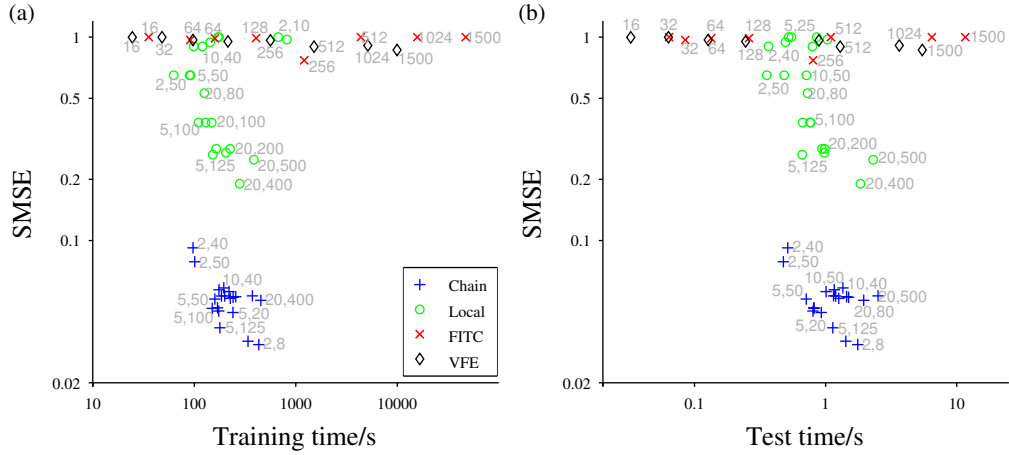

Figure 3: Experiment 2. Filtered audio signal reconstruction error as a function of training time (a) and test time (b) for different approximations. See caption of fig. 2 for full details.

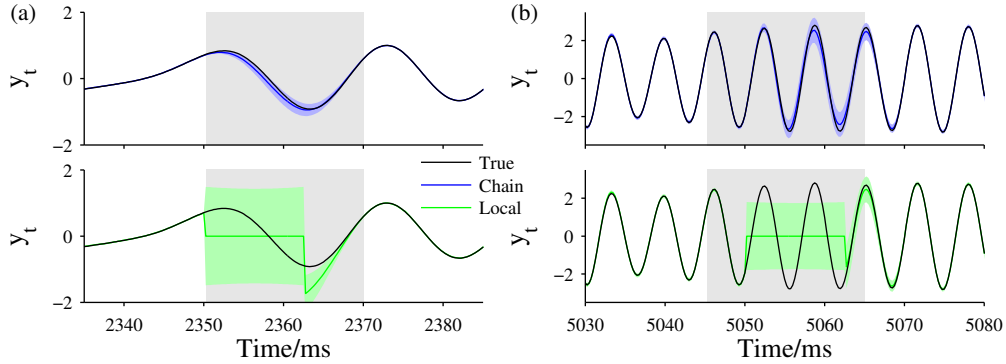

Figure 4: Missing data imputation for experiment 1 (audio sub-band data, (a)) and experiment 2 (filtered audio data, (b)). Imputation using the chain-structured approximation (top) is more accurate and less uncertain than the predictions obtained from the local method (bottom). Blocks consisted of 5 pseudo-datapoints and 50 observations respectively.

pseudo-point approximation methods considered in the first experiment. Figure 5 shows the speed-accuracy trade-off for the various approximation methods at the test and training stages. We found that the global approximation techniques such as FITC or SSGP could not tractably handle a sufficient number of pseudo-datapoints to support accurate imputation. The local variant of our method outperformed the other techniques, but compared poorly to the tree. Typical reconstructions from the tree, local and FITC approximations are shown in fig. 6.

**Summary of experimental results** The speed-accuracy frontier for the new approximation scheme dominates those produced by the other methods over a wide range for each of the three datasets. Similar results were found for additional datasets (see supplementary material). It is perhaps not surprising that the tree approximation performs so favourably. Consider the rule-of-thumb estimate for the number of pseudo-datapoints required. Using the length-scales $l_d$ learned by the tree-approximation as a proxy for the posterior dependency length the estimated pseudo-dataset size required for the three datasets is $M \gtrsim \prod_d L_d/l_d \approx \{1400, 1000, 5000\}$. This is at the upper end of what can be tractably handled using standard approximations. Moreover, these approximation schemes can be made arbitrarily poor by expanding the region further. The most accurate tree-structured approximation for the three datasets used $\{2500, 10000, 20000\}$ datapoints respectively. The local PIC method performs more favourably than the standard approximations and is generally faster than the tree since it involves a single pass through the dataset and simpler matrix computations. However, blocking the data into independent chunks results in artifacts at the block boundaries which reduces the approximation's accuracy significantly when compared to the tree (e.g. if they happen to coincide with a missing region).

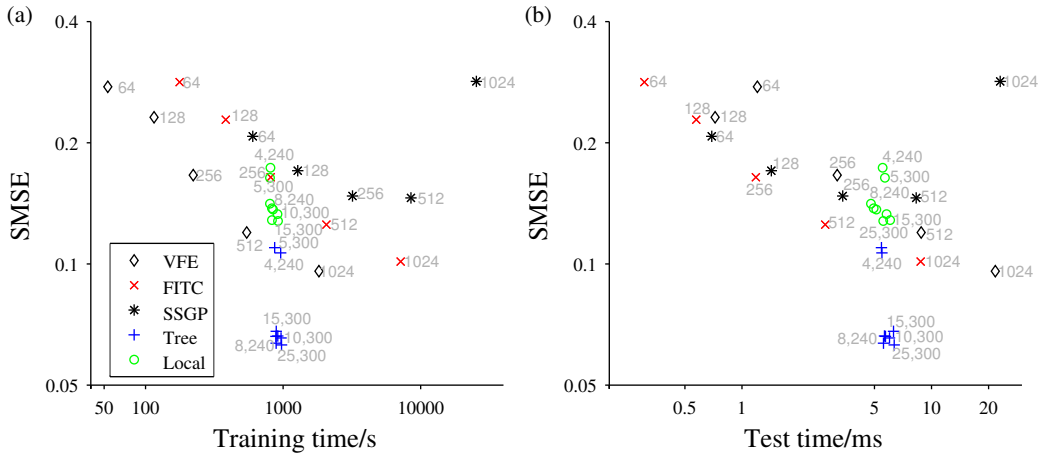

Figure 5: Experiment 3. Terrain data reconstruction. SMSE as a function of training time (a) and test time (b). See caption of fig. 2 for full details.

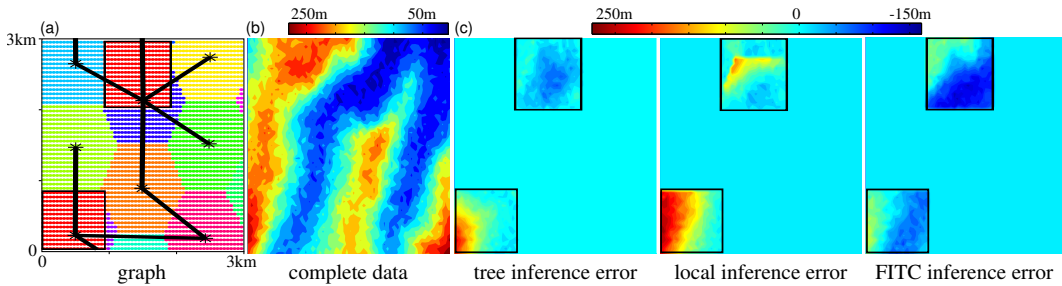

Figure 6: Experiment 3. Terrain data reconstruction. The blocks in this region input space are organized into a tree-structure (a) with missing regions shown by the black squares. The complete terrain altitude data for the region (b). Prediction errors from three methods (c).

## 4   Conclusion

This paper has presented a new pseudo-datapoint approximation scheme for Gaussian process regression problems which imposes a tree or chain structure on the pseudo-dataset that is calibrated using a KL divergence. Inference and learning in the resulting approximate model proceeds efficiently via Gaussian belief propagation. The computational cost of the approximation is linear in the pseudo-dataset size, improving upon the quadratic scaling of typical approaches, and opening the door to more challenging datasets than have previously been considered. Importantly, the method does not require the input data or the covariance function to have special structure (stationarity, regular sampling, time-series settings etc. are *not* a requirement). We showed that the approximation obtained a superior performance in both predictive accuracy and runtime complexity on challenging regression tasks which included audio missing data imputation and spatial terrain prediction.

There are several directions for future work. First, the new approximation scheme should be tested on datasets that have higher dimensional input spaces since it is not clear how well the approximation will generalize to this setting. Second, the tree structure naturally leads to (possibly distributed) online stochastic inference procedures in which gradients computed at a local block, or a collection of local blocks, are used to update hyperparameters directly, as opposed waiting for a full pass up and down the tree. Third, the tree structure used for prediction can be decoupled from the tree structure used for training, whilst still employing the same pseudo-datapoints potentially improving prediction.

### Acknowledgements

We would like to thank the EPSRC (grant numbers EP/G050821/1 and EP/L000776/1) and Google for funding.

## Footnotes

[1]Here and in what follows, the dependence on the input values $\mathbf{x}$ has been suppressed to lighten the notation.

[2]Synthetic data experiments can be found in the supplementary material.

[3]Code is available at `http://www.gaussianprocess.org/gpml/code/matlab/doc/` [FITC], `http://www.tsc.uc3m.es/~miguel/downloads.php` [SSGP], `http://becs.aalto.fi/en/research/bayes/gpstuff/` [SDE] and `http://mlg.eng.cam.ac.uk/thang/` [Tree+VFE].

[4]Dataset is available at `http://data.gov.uk/dataset/os-terrain-50-dtm`.

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
