[Supplementary Material]

# Tree-structured Gaussian process approximations
# Supplementary material

**Thang Bui**
tdb40@cam.ac.uk

**Richard Turner**
ret26@cam.ac.uk

Computational and Biological Learning Lab, Department of Engineering
University of Cambridge, Trumpington Street, Cambridge, CB2 1PZ, UK

## 1 KL justification for different approximations

### 1.1 The Fully Independent Training Conditional (FITC) approximation

The FITC approximation assumes that all latent functions $\mathbf{f}$ are independent given inducing variables $\mathbf{u}$. The joint distribution of the inducing variables and the latent functions $q(\mathbf{f}, \mathbf{u})$ is then chosen by minimising the KL divergence between $p(\mathbf{f}, \mathbf{u})$ and $q(\mathbf{f}, \mathbf{u})$,

$$q(f_i|\mathbf{u}) \leftarrow \underset{q(f_i|\mathbf{u})}{\arg\min} \ \mathrm{KL}(p(\mathbf{f}, \mathbf{u})||q(\mathbf{f}, \mathbf{u})), \tag{1}$$

subject to $q(\mathbf{f}|\mathbf{u}) = \prod_i q(f_i|\mathbf{u})$ and $\int \mathrm{d}f_i q(f_i|\mathbf{u}) = 1$. It is noted that $\mathrm{KL}(a||b)$ is the measurement of information "lost" when using $b$ to approximate $a$. It was argued in [1] that it is appropriate to use this KL divergence as an approximation measure since we are trying to find a sparse representation $\mathbf{u}$ and its relationship with $\mathbf{f}$ to approximate $p$ by $q$.

The KL divergence above can be expanded as follows,

$$\mathcal{L}_1 = \mathrm{KL}(p(\mathbf{f}, \mathbf{u})||q(\mathbf{f}, \mathbf{u})) \tag{2}$$

$$= \mathcal{C} - \int \mathrm{d}\mathbf{u} \, \mathrm{d}\mathbf{f} p(\mathbf{f}, \mathbf{u}) \log q(\mathbf{f}|\mathbf{u}) \tag{3}$$

$$= \mathcal{C} - \int \mathrm{d}\mathbf{u} \, \mathrm{d}\mathbf{f} p(\mathbf{f}, \mathbf{u}) \sum_i \log q(f_i|\mathbf{u}) \tag{4}$$

$$= \mathcal{C} - \sum_i \int \mathrm{d}\mathbf{u} \, \mathrm{d}\mathbf{f} p(\mathbf{f}, \mathbf{u}) \log q(f_i|\mathbf{u}), \tag{5}$$

where $\mathcal{C}$ is a constant w.r.t $q(\mathbf{f}|\mathbf{u})$. Adding the Lagrange multipliers into the above expression and taking the derivative w.r.t. $q(f_i|\mathbf{u})$ give us,

$$\frac{\partial}{\partial q(f_i|\mathbf{u})} \mathcal{L}_1 = \int \mathrm{d}\mathbf{f}_{\neq i} p(\mathbf{f}, \mathbf{u}) \frac{1}{q(f_i|\mathbf{u})} \tag{6}$$

$$= -\frac{p(f_i, \mathbf{u})}{q(f_i|\mathbf{u})} + \lambda_i \tag{7}$$

By setting the derivative in 7 to zero and applying the normalisation constraint for $q(f_i|\mathbf{u})$ we have,

$$\boxed{q(f_i|\mathbf{u}) = \frac{p(f_i, \mathbf{u})}{p(\mathbf{u})} = p(f_i|\mathbf{u})}. \tag{8}$$

**Extension to the Partially Independent Training Conditional (PITC) approximation**

When the dependency assumption between latent functions given the inducing variables is relaxed, or $q(\mathbf{f}|\mathbf{u}) = \prod_i q(\mathbf{f}_i|\mathbf{u})$ where $\mathbf{f}_i$ is a cluster of latent functions, by using the same derivation as

above, we obtain:

$$q(\mathbf{f}_i|\mathbf{u}) = \frac{p(\mathbf{f}_i, \mathbf{u})}{p(\mathbf{u})} = p(\mathbf{f}_i|\mathbf{u}), \tag{9}$$

which is exactly the same as in FITC but $f_i$ is now replaced by $\mathbf{f}_i$.

**Extension to tree-structured inducing inputs**

When the inducing inputs $\mathbf{u}$ is tree-structured, the forward KL divergence can be written as,

$$\begin{aligned}
\mathcal{L}_2 &= \mathrm{KL}(p(\mathbf{f}, \mathbf{u})||q(\mathbf{f}, \mathbf{u})) \tag{10} \\
&= \mathcal{C} - \int \mathrm{d}\mathbf{u}\, \mathrm{d}\mathbf{f} p(\mathbf{f}, \mathbf{u}) \log[q(\mathbf{f}|\mathbf{u}_i)q(\mathbf{u})] \tag{11} \\
&= \mathcal{C} - \int \mathrm{d}\mathbf{u}\, \mathrm{d}\mathbf{f} p(\mathbf{f}, \mathbf{u}) \log[\prod_i q(\mathbf{f}_i|\mathbf{u}_i) \prod_{t=1}^{T} q(\mathbf{u}_t|\mathbf{u}_{pt})] \quad (pt \text{ is the parent node of } t) \tag{12} \\
&= \mathcal{C} - \sum_i \int \mathrm{d}\mathbf{u}\, \mathrm{d}\mathbf{f} p(\mathbf{f}, \mathbf{u}) \log q(\mathbf{f}_i|\mathbf{u}) - \sum_t \int \mathrm{d}\mathbf{u}\, \mathrm{d}\mathbf{f} p(\mathbf{f}, \mathbf{u}) \log q(\mathbf{u}_t|\mathbf{u}_{pt}). \tag{13}
\end{aligned}$$

By using the normalisation constraints for $q(\mathbf{u}_t|\mathbf{u}_{pt})$ and $q(\mathbf{f}_i|\mathbf{u}_i)$, the Lagrange multipliers can be added to find the optimal conditionals that minimize the KL divergence. The derivative of the Lagrangian w.r.t. $q(\mathbf{f}_i|\mathbf{u})$ is,

$$\frac{\partial}{\partial q(\mathbf{f}_i|\mathbf{u}_i)}\mathcal{L}_2 = -\frac{p(\mathbf{f}_i, \mathbf{u}_i)}{q(\mathbf{f}_i|\mathbf{u}_i)} + \lambda_i. \tag{14}$$

As above, by equating this derivative to zero and applying the normalisation constraint, we have,

$$q(\mathbf{f}_i|\mathbf{u}) = \frac{p(\mathbf{f}_i, \mathbf{u}_i)}{p(\mathbf{u}_i)} = p(\mathbf{f}_i|\mathbf{u}_i). \tag{15}$$

The derivative of the Lagrangian w.r.t. $q(\mathbf{u}_t|\mathbf{u}_{pt})$ is,

$$\begin{aligned}
\frac{\partial}{\partial q(\mathbf{u}_t|\mathbf{u}_{pt})}\mathcal{L}_2 &= -\frac{\int \mathrm{d}\mathbf{f} p(\mathbf{f}, \mathbf{u}_t, \mathbf{u}_{pt})}{q(\mathbf{u}_t|\mathbf{u}_{pt})} + \lambda_t \tag{16} \\
&= \frac{p(\mathbf{u}_t, \mathbf{u}_{pt})}{q(\mathbf{u}_t|\mathbf{u}_{pt})} + \lambda_t \tag{17}
\end{aligned}$$

Similarly, using the normalisation constraint gives us $\lambda_t = p(\mathbf{u}_{pt})$ which leads to,

$$q(\mathbf{u}_t|\mathbf{u}_{pt}) = \frac{p(\mathbf{u}_t, \mathbf{u}_{pt})}{p(\mathbf{u}_{pt})} = p(\mathbf{u}_t|\mathbf{u}_{pt}). \tag{18}$$

## 1.2 The Deterministic Training Conditional (DTC) approximation

The likelihood approximation presented in [2, 3] can be justified by choosing a likelihood function $q(\mathbf{y}|\mathbf{u})$ to minimize the KL divergence,

$$q(\mathbf{y}|\mathbf{u}) \leftarrow \underset{q(\mathbf{y}|\mathbf{u})}{\arg\min}\ \mathrm{KL}(q(\mathbf{f}, \mathbf{u}|\mathbf{y})||p(\mathbf{f}, \mathbf{u}|\mathbf{y})), \tag{19}$$

where,

$$q(\mathbf{f}, \mathbf{u}|\mathbf{y}) = \frac{q(\mathbf{y}|\mathbf{u})p(\mathbf{f}|\mathbf{u})p(\mathbf{u})}{q(\mathbf{y})}, \tag{20}$$

$$p(\mathbf{f}, \mathbf{u}|\mathbf{y}) = \frac{p(\mathbf{y}|\mathbf{f})p(\mathbf{f}|\mathbf{u})p(\mathbf{u})}{p(\mathbf{y})}, \tag{21}$$

$$\text{and,} \quad q(\mathbf{y}) = \int \mathrm{d}\mathbf{u} p(\mathbf{u})p(\mathbf{y}|\mathbf{u}). \tag{22}$$

Consider the likelihood $q(\mathbf{y}|\mathbf{u})$ that is a valid distribution or $\int \mathrm{d}y q(\mathbf{y}|\mathbf{u}) = 1$ [4], combining the reversed KL divergence and the normalisation constraint gives us the Lagrangian:

$$\mathcal{L}_3 = \mathrm{KL}(q(\mathbf{f}, \mathbf{u}|\mathbf{y})||p(\mathbf{f}, \mathbf{u}|\mathbf{y})) + \lambda \left( \int \mathrm{d}\mathbf{y} q(\mathbf{y}|\mathbf{u}) - 1 \right) \tag{23}$$

$$= \log p(\mathbf{y}) - \log q(\mathbf{y}) + \int \mathrm{d}\mathbf{f}\,\mathrm{d}\mathbf{u} \frac{q(\mathbf{y}|\mathbf{u})p(\mathbf{f}|\mathbf{u})p(\mathbf{u})}{q(\mathbf{y})} \log \frac{q(\mathbf{y}|\mathbf{u})}{p(\mathbf{y}|\mathbf{f})} + \lambda \left( \int \mathrm{d}\mathbf{y} q(\mathbf{y}|\mathbf{u}) - 1 \right). \tag{24}$$

The derivative $\mathcal{L}_3$ w.r.t $q(\mathbf{y}|\mathbf{u})$ is,

$$\frac{\partial}{\partial q(\mathbf{y}|\mathbf{u})} \mathcal{L}_3 = \frac{1}{q(\mathbf{y})} p(\mathbf{u}) + \int \mathrm{d}\mathbf{f} \frac{p(\mathbf{f}|\mathbf{u})p(\mathbf{u})}{q(\mathbf{y})} \log \frac{q(\mathbf{y}|\mathbf{u})}{p(\mathbf{y}|\mathbf{f})} + \int \mathrm{d}\mathbf{f} \frac{q(\mathbf{y}|\mathbf{u})p(\mathbf{f}|\mathbf{u})p(\mathbf{u})}{q(\mathbf{y})} \frac{1}{q(\mathbf{y}|\mathbf{u})} + \lambda \tag{25}$$

$$= \frac{p(\mathbf{u})}{q(\mathbf{y})} \left[ \log q(\mathbf{y}|\mathbf{u}) - \int \mathrm{d}\mathbf{f} p(\mathbf{f}|\mathbf{u}) \log p(\mathbf{y}|\mathbf{f}) \right] + \lambda. \tag{26}$$

Setting 26 to zero gives,

$$q(\mathbf{y}|\mathbf{u}) = \exp \left( -\frac{\lambda q(\mathbf{y})}{p(\mathbf{u})} \right) \exp \left( \int \mathrm{d}\mathbf{f} p(\mathbf{f}|\mathbf{u}) \log p(\mathbf{y}|\mathbf{f}) \right). \tag{27}$$

The integral inside the exponential above can be computed analytically as follows,

$$\mathrm{M} = \int \mathrm{d}\mathbf{f} p(\mathbf{f}|\mathbf{u}) \log p(\mathbf{y}|\mathbf{f}) \tag{28}$$

$$= \int \mathrm{d}\mathbf{f} \mathcal{N}(\mathbf{f}; K_{\mathbf{fu}}K_{\mathbf{uu}}^{-1}\mathbf{u}, K_{\mathbf{ff}} - K_{\mathbf{fu}}K_{\mathbf{uu}}^{-1}K_{\mathbf{uf}}) \log[\mathcal{N}(\mathbf{y}; \mathbf{f}, \sigma^2 I)] \tag{29}$$

$$= \int \mathrm{d}\mathbf{f} \mathcal{N}(\mathbf{f}; \mathbf{A}, \mathbf{B}) \left[ -\frac{n}{2} \log(2\pi\sigma^2) - \frac{1}{2\sigma^2}(\mathbf{y} - \mathbf{f})^{\mathsf{T}} I (\mathbf{y} - \mathbf{f}) \right] \tag{30}$$

$$= \int \mathrm{d}\mathbf{f} \mathcal{N}(\mathbf{f}; \mathbf{A}, \mathbf{B}) \left[ -\frac{n}{2} \log(2\pi\sigma^2) - \frac{1}{2\sigma^2} \mathrm{Tr}(\mathbf{y}\mathbf{y}^{\mathsf{T}} - 2\mathbf{y}\mathbf{f}^{\mathsf{T}} + \mathbf{f}\mathbf{f}^{\mathsf{T}}) \right] \tag{31}$$

$$= -\frac{n}{2} \log(2\pi\sigma^2) - \frac{1}{2\sigma^2} \mathrm{Tr}(\mathbf{y}\mathbf{y}^{\mathsf{T}} - 2\mathbf{y}\mathbf{A}^{\mathsf{T}} + \mathbf{A}\mathbf{A}^{\mathsf{T}} + \mathbf{B}) \tag{32}$$

$$= -\frac{1}{2\sigma^2} \mathrm{Tr}(\mathbf{B}) + \log[\mathcal{N}(\mathbf{y}; \mathbf{A}, \sigma^2 I)], \tag{33}$$

where $\mathbf{A} = K_{\mathbf{fu}}K_{\mathbf{uu}}^{-1}\mathbf{u}$ and $\mathbf{B} = K_{\mathbf{ff}} - K_{\mathbf{fu}}K_{\mathbf{uu}}^{-1}K_{\mathbf{uf}}$. Substitute 33 into 27 and use the normalisation constraint, the optimal form for the approximate likelihood is:

$$\boxed{q(\mathbf{y}|\mathbf{u}) = \mathcal{N}(\mathbf{y}; K_{\mathbf{fu}}K_{\mathbf{uu}}^{-1}\mathbf{u}, \sigma^2 I)} \tag{34}$$

As noted in [1], the same result can be obtained by optimising the KL divergence between the joint models of $\mathbf{y}$, $\mathbf{f}$ and $\mathbf{u}$: $\mathrm{KL}(q(\mathbf{y}|\mathbf{u})p(\mathbf{f}|\mathbf{u})p(\mathbf{u})||p(\mathbf{y}|\mathbf{f})p(\mathbf{f}|\mathbf{u})p(\mathbf{u}))$.

Let's remove the normalisation constraint of the likelihood term, this equivalently means that the Lagrange multiplier in equation 27 is zero, or the optimal likelihood is:

$$\boxed{q(\mathbf{y}|\mathbf{u}) = \exp \left( -\frac{1}{2\sigma^2} \mathrm{Tr}(K_{\mathbf{ff}} - K_{\mathbf{fu}}K_{\mathbf{uu}}^{-1}K_{\mathbf{uf}}) \right) \mathcal{N}(\mathbf{y}; K_{\mathbf{fu}}K_{\mathbf{uu}}^{-1}\mathbf{u}, \sigma^2 I)} \tag{35}$$

Here it becomes clear that why the expression of the posterior of $\mathbf{u}$ in DTC is exactly the same as in [5], only the approximate marginal likelihood is modified. Both are optimising the same KL divergence under the approximate likelihood regime, but [5] allows a free form for the likelihood (which turns out to be easily computed analytically) as opposed to a Gaussian likelihood in [4].

## 2 Relationship between our method and previous approximations

Table 1 shows the relationship between the tree-structured prior approximation scheme and previous works.

| model | number of blocks | pseudo-datapoints number (location) | parameters intra-block. | inter-block |
|---|---|---|---|---|
| full Gaussian Process (GP) | $K = 1$ | $M = N$ (at data inputs) | full | N/A |
| PITC | $K = 1$ | $M < N$ | blkdiag$[\mathbf{R}_k]$ | N/A |
| FITC | $K = 1$ | $M < N$ | diag$[\mathbf{R}_k]$ | N/A |
| local | many | $M = N$ (at data inputs) | full | $\mathbf{A}_k = \mathbf{0}$ and $\mathbf{Q}_k = \mathbf{K}_{\mathbf{u}_k \mathbf{u}_k}$ |
| local PITC | many | $M < N$ | full | $\mathbf{A}_k = \mathbf{0}$ and $\mathbf{Q}_k = \mathbf{K}_{\mathbf{u}_k \mathbf{u}_k}$ |
| local FITC | many | $M < N$ | diag$[\mathbf{R}_k]$ | $\mathbf{A}_k = \mathbf{0}$ and $\mathbf{Q}_k = \mathbf{K}_{\mathbf{u}_k \mathbf{u}_k}$ |

Table 1: Relationship between the tree-structured prior approximation scheme and other prior approximation methods. diag$[\mathbf{A}]$ and blkdiag$[\mathbf{A}]$ are shorthand for diagonal and block-diagonal versions of the matrix $\mathbf{A}$

## 3 Spectral mixture kernel approximation

Consider a function that is drawn from a GP with the following covariance function,

$$k(t_1, t_2) = \sum_{m=1}^{M} \sigma_m^2 \cos(\omega_m(t_1 - t_2)) \exp\left(-\frac{1}{2l_m^2}(t_1 - t_2)^2\right), \qquad (36)$$

which is a sum of products of a periodic sinunoid and a squared exponential covariance functions where $\sigma_m^2, l_m$ and $\omega_m$ are the signal variance, characteristic lengthscale and frequency of the spectral component $m$ respectively. The observations are the function values corrupted by i.i.d, Gaussian noise of variance $\sigma_n^2$. In this section, we only consider one dimensional, regularly spaced input. The approximate model with chain-structured inducing inputs can be described graphically as follows,

Figure 1: Chain-structured inducing-input approximation

The generative model can be written as follows,

$$p(\mathbf{u}|\theta) = \prod_{j=1}^{K} p(\mathbf{u}_j|\mathbf{u}_{j-1}, \theta) = \prod_{j=1}^{K} \mathcal{N}(\mathbf{u}_j|\mathbf{A}\mathbf{u}_{j-1}, \mathbf{Q}), \qquad (37)$$

$$p(\mathbf{f}|\mathbf{u}) = \prod_{j=1}^{K} p(\mathbf{f}_j|\mathbf{u}_j) = \prod_{j=1}^{K} \mathcal{N}(\mathbf{f}_j|\mathbf{C}_j\mathbf{u}_j, \mathbf{R}_j), \qquad (38)$$

$$p(\mathbf{y}|\mathbf{f}) = \prod_{i=1}^{n} p(y_i; f_i, \sigma_n^2), \qquad (39)$$

where

$$\mathbf{A} = \mathrm{diag}(\begin{bmatrix} A^1 & A^1 & A^2 & A^2 & \cdots & A^M & A^M \end{bmatrix}),$$
$$\mathbf{Q} = \mathrm{diag}(\begin{bmatrix} Q^1 & Q^1 & Q^2 & Q^2 & \cdots & Q^M & Q^M \end{bmatrix}),$$
$$\mathbf{C}_j = \begin{bmatrix} W_{j1}^1 C^1 & W_{j2}^1 C^1 & W_{j1}^2 C^2 & W_{j2}^2 C^2 & \cdots & W_{j1}^M C^1 & W_{j2}^M C^M \end{bmatrix},$$
$$\mathbf{R}_j = \mathcal{W}\mathcal{R}\mathcal{W}^\mathsf{T},$$

$$
\begin{aligned}
\mathcal{R} &= \mathrm{diag}(\begin{bmatrix} R^1 & R^1 & R^2 & R^2 & \cdots & R^M & R^M \end{bmatrix}), \\
\mathcal{W} &= \begin{bmatrix} W_{j1}^1 & W_{j2}^1 & W_{j1}^2 & W_{j2}^2 & \cdots & W_{j1}^M & W_{j2}^M \end{bmatrix}, \\
A^m &= K_{\mathbf{u}\mathbf{u}_p}^m K_{\mathbf{u}_m \mathbf{u}_m}^{m,-1}, \\
Q^m &= K_{\mathbf{u}\mathbf{u}}^m - K_{\mathbf{u}\mathbf{u}_p}^m K_{\mathbf{u}_p \mathbf{u}_p}^{m,-1} K_{\mathbf{u}_p \mathbf{u}}^m, \\
C^m &= K_{\mathbf{f}\mathbf{u}}^m K_{\mathbf{u}\mathbf{u}}^{m,-1}, \\
R^m &= K_{\mathbf{f}\mathbf{f}}^m - K_{\mathbf{f}\mathbf{u}}^m K_{\mathbf{u}\mathbf{u}}^{m,-1} K_{\mathbf{u}\mathbf{f}}^m, \\
W_{j1}^m &= \mathrm{diag}[\cos(\omega_m t_{p+1}), \cos(\omega_m t_{p+2}), \cdots \cos(\omega_m t_{p+\tau_1})] \\
W_{j2}^m &= \mathrm{diag}[\sin(\omega_m t_{p+1}), \sin(\omega_m t_{p+2}), \cdots \sin(\omega_m t_{p+\tau_1})],
\end{aligned}
$$

$\tau_1$ and $\tau_2$ are the number of observations and inducing inputs per clusters respectively, $t_p = j\tau_1, A^m \in \mathbb{R}^{\tau_2} \times \mathbb{R}^{\tau_2}, Q^m \in \mathbb{R}^{\tau_2} \times \mathbb{R}^{\tau_2}, C^m \in \mathbb{R}^{\tau_1} \times \mathbb{R}^{\tau_2}, R^m \in \mathbb{R}^{\tau_1} \times \mathbb{R}^{\tau_1}, \{W_{j1}, W_{j2}\} \in \mathbb{R}^{\tau_1} \times \mathbb{R}^{\tau_1}, C_j \in \mathbb{R}^{\tau_1} \times \mathbb{R}^{2M\tau_2}, R_j \in \mathbb{R}^{\tau_1} \times \mathbb{R}^{\tau_1}, K^m$ is the covariance matrix of the exponentiated quadratic component $m$ with lengthscale $l_m$ and signal variance $\sigma_m^2$, and $\mathrm{diag}$ is an operator that takes the vector of numbers or matrices and forms the corresponding diagonal or block diagonal matrix.

## 4  Gaussian belief propagation in tree/chain graphical models

Let $x$ and $y$ be the hidden variables and observations respectively, $y_t$ be the observations emitted by hidden variables $x_t$. It is noted that all variables in this section are in vector forms and we have omitted the boldface notation as in the previous sections. Given the transition and emission parameters, the Gaussian belief propagation algorithm can be used to infer the marginal distribution of the latent variables given the observations. Furthermore, since the marginal likelihood of the parameters given the observations can be found, it can be optimized to obtain the ML parameters for our model. We describe here the message passing scheme to compute the marginal likelihood and its derivative w.r.t the model parameters.

### 4.1  Message passing

The joint posterior distribution of $\mathbf{u}$ is

$$
p(\mathbf{u}|\mathbf{y}) \propto \prod_{i \in \mathcal{V}} \exp\left(-\frac{1}{2}\mathbf{u}_i^\mathsf{T} \mathbf{J}_i \mathbf{u}_i + \mathbf{u}_i^\mathsf{T} \mathbf{h}_i\right) \prod_{i,j \in \mathcal{E}} \exp\left(\mathbf{u}_i^\mathsf{T} \mathbf{J}_{ij} \mathbf{u}_j\right) \tag{40}
$$

where $\mathcal{V}$ and $\mathcal{E}$ denote vertex and edge sets accordingly, $\mathbf{J}_i = \mathbf{J}_{ii} + \mathbf{C}_i^\mathsf{T}(\mathbf{R}_i + \sigma^2\mathbf{I}_i)^{-1}\mathbf{C}_i + \sum_{j \in \mathrm{nei}(i)} \mathbf{A}_j^\mathsf{T} \mathbf{Q}_j^{-1} \mathbf{A}_j$ with $\mathbf{J}_{ii}$ being the initial precision matrix $\mathbf{P}_o^{-1}$ for the root node and $\mathbf{Q}_i^{-1}$ otherwise, $\mathbf{h}_i = \mathbf{C}_i^\mathsf{T} \mathbf{R}_i^{-1} \mathbf{y_i}$, and $\mathbf{J}_{ij} = \mathbf{Q}_i^{-1}\mathbf{A}_i$.

### 4.2  Cross covariance

We give the details to compute an analytic form for the cross covariance which is need for the derivative. Consider the following simple graphical model where $y_1$ includes observations 'emitted' by $x_1$ and 'on the left' of $x_1$, $y_2$ includes observations 'emitted' by $x_2$ and 'on the right' of $x_2$, $A$ and $Q$ are the transition dynamics connecting $x_1$ and $x_2$ such that $p(x_2|x_1) = \mathcal{N}(x_2; Ax_1, Q)$.

By using the Gaussian belief propagation algorithm as discussed above, one can obtain $p(x_1|y_1), p(x_2|y_2)$ and the marginal distributions $p(x_1|y_1, y_2)$ and $p(x_2|y_1, y_2)$. Assume that

$p(x_1|y_1) = \mathcal{N}^{-1}(x_1; h_1, J_1)$ and $p(x_2|y_1, y_2) = \mathcal{N}^{-1}(x_2; h_2, J_2)$ ($\mathcal{N}^{-1}$ denotes a normal distribution in canonical form), we have,

$$p(x_1, x_2|y_1) = p(x_2|x_1)p(x_1|y_1) \tag{41}$$

$$\propto \exp\left(\frac{1}{2}(x_2 - Ax_1)^\intercal Q^{-1}(x_2 - Ax_1)\right)\exp(-\frac{1}{2}x_1^\intercal J_1 x_1 + h_1^\intercal x_1) \tag{42}$$

$$\propto \exp\left(-\frac{1}{2}\begin{bmatrix} x_2 \\ x_1 \end{bmatrix}^\intercal \begin{bmatrix} Q^{-1} & -Q^{-1}A \\ -A^\intercal Q^{-1} & J_1 + A^\intercal Q^{-1}A \end{bmatrix}\begin{bmatrix} x_2 \\ x_1 \end{bmatrix} + \begin{bmatrix} 0 \\ h_1 \end{bmatrix}^\intercal \begin{bmatrix} x_2 \\ x_1 \end{bmatrix}\right). \tag{43}$$

Marginalising out $x_1$ gives

$$p(x_2|y_1) \propto \exp\left(-\frac{1}{2}x_2^\intercal(Q + AJ_1^{-1}A^\intercal)^{-1}x_2 + [Q^{-1}A(J_1 + A^\intercal Q^{-1}A)h_1]^\intercal x_2\right). \tag{44}$$

Therefore,

$$p(x_1, x_2|y_1, y_2) = p(x_1|x_2, y_1)p(x_2|y_1, y_2) \tag{45}$$

$$= p(x_1, x_2|y_1)p(x_2|y_1, y_2)/p(x_2|y_1) \tag{46}$$

$$\propto \exp\left(\frac{1}{2}(x_2 - Ax_1)^\intercal Q^{-1}(x_2 - Ax_1)\right)\exp(-\frac{1}{2}x_1^\intercal J_1 x_1 + h_1^\intercal x_1)$$

$$\exp\left(\frac{1}{2}x_2^\intercal(Q + AJ_1^{-1}A^\intercal)^{-1}x_2 - [Q^{-1}A(J_1 + A^\intercal Q^{-1}A)h_1]^\intercal x_2\right)$$

$$\exp(-\frac{1}{2}x_2^\intercal J_2 x_2 + h_2^\intercal x_2) \tag{47}$$

$$\propto \exp\left(-\frac{1}{2}\begin{bmatrix} x_1 \\ x_2 \end{bmatrix}^\intercal \begin{bmatrix} J_1 + A^\intercal Q^{-1}A & -A^\intercal Q^{-1} \\ -Q^{-1}A & J_2 + Q^{-1}A(J_1 + A^\intercal Q^{-1}A)A^\intercal Q^{-1} \end{bmatrix}\begin{bmatrix} x_1 \\ x_2 \end{bmatrix}\right)$$

$$\exp\left(\begin{bmatrix} h_1 \\ h_2 + Q^{-1}A(J_1 + A^\intercal Q^{-1}A)h_1 \end{bmatrix}^\intercal \begin{bmatrix} x_1 \\ x_2 \end{bmatrix}\right). \tag{48}$$

By inverting the precision matrix in the equation above, we get the cross covariance matrix between $x_1$ and $x_2$,

$$V_{12} = (J_1 + A^\intercal Q^{-1}A)^{-1}A^\intercal Q^{-1}J_2^{-1}. \tag{49}$$

### 4.3 Log marginal likelihood and its derivatives

#### 4.3.1 Message passing to compute the log marginal likelihood

We will consider a small example and show how to use message passing in trees to compute the log marginal likelihood. Consider a graph as in figure 2 where only observation nodes are shown for clarity.

Figure 2: Illustrative example for computing the log marginal likelihood using message passing

By using the rules of probability, we can write the likelihood of the parameter given the data:

$$p(y_{1:9}) = p(y_9|y_{1:8})p(y_{1:8}) \tag{50}$$

$$= p(y_9|y_{1:8})p(y_6|y_{1:5,7,8})p(y_{1:5,7,8}) \tag{51}$$

$$= p(y_9|y_{1:8})p(y_6|y_{1:5,7,8})p(y_7|y_{1:5,8})p(y_{1:5,8}) \tag{52}$$

$$= p(y_9|y_{1:8})p(y_6|y_{1:5,7,8})p(y_7|y_{1:5,8})p(y_8|y_{1:5})p(y_{1:5}) \tag{53}$$

$$= p(y_9|y_{1:8})p(y_6|y_{1:5,7,8})p(y_7|y_{1:5,8})p(y_8|y_{1:5})p(y_3|y_{1,2,4,5})p(y_{1,2,4,5}) \tag{54}$$

$$= p(y_9|y_{1:8})p(y_6|y_{1:5,7,8})p(y_7|y_{1:5,8})p(y_8|y_{1:5})p(y_3|y_{1,2,4,5})p(y_4|y_{1,2,5})p(y_{1,2,5}) \tag{55}$$

$$= p(y_9|y_{1:8})p(y_6|y_{1:5,7,8})p(y_7|y_{1:5,8})p(y_8|y_{1:5})p(y_3|y_{1,2,4,5})p(y_4|y_{1,2,5})p(y_5|vy_{1,2})p(y_{1,2}) \tag{56}$$

$$= p(y_9|y_{1:8})p(y_6|y_{1:5,7,8})p(y_7|y_{1:5,8})p(y_8|y_{1:5})p(y_3|y_{1,2,4,5})p(y_4|y_{1,2,5})p(y_5|y_{1,2})p(y_2|y_1)p(y_1) \tag{57}$$

By comparing the nodes that are children of a node (e.g.: nodes 6, 7, 8 are children of 3), the term each contributes to the marginal likelihood looks slightly different in the variables they condition on: the first child conditions on all other children, the second child conditions on all other children except the first child and so on. We can make use of this pattern to derive a message passing scheme to compute the marginal likelihood.

**Data**: Observations
**Result**: Log marginal likelihood
Initialize root message $m_{par_{root}} = \emptyset$;
**for** *all nodes at each layer in the tree* **do**
    compute local likelihood term using its observation $y_j$ and parent message $m_{par_j}$;
    **if** *there is a child* **then**
        combine local observations and all children messages $m_i$ to form $M$;
        **for** *all childrens* **do**
            subtract child message $m_i$ from $M$ to form $M^{\backslash i}$;
            $M \leftarrow M^{\backslash i}$;
            pass $M$ to the child $i$: $m_{par_i} = M$;
        **end**
    **end**
**end**

**Algorithm 1:** Message passing algorithm to compute log marginal likelihood

### 4.3.2 The derivative of the log marginal likelihood

The log marginal likelihood of the parameters given the observations is $\mathcal{L} = \log p(\mathbf{y}|\theta)$. Its derivative w.r.t the parameter $\theta$ is,

$$\frac{\mathrm{d}}{\mathrm{d}\theta}\mathcal{L} = \frac{1}{p(\mathbf{y}|\theta)}\frac{\mathrm{d}}{\mathrm{d}\theta}p(\mathbf{y}|\theta) \tag{58}$$

$$= \frac{1}{p(\mathbf{y}|\theta)}\frac{\mathrm{d}}{\mathrm{d}\theta}\int \mathrm{d}x\, p(y,x|\theta) \tag{59}$$

$$= \int \mathrm{d}x\, \frac{1}{p(\mathbf{y}|\theta)}\frac{\mathrm{d}}{\mathrm{d}\theta}p(y,x|\theta) \tag{60}$$

$$= \int \mathrm{d}x\, \frac{1}{p(\mathbf{y}|\theta)}p(y,x|\theta)\frac{\mathrm{d}}{\mathrm{d}\theta}\log p(y,x|\theta) \tag{61}$$

$$= \int \mathrm{d}x\, p(x|y,\theta)\frac{\mathrm{d}}{\mathrm{d}\theta}\log\left[p(x_1|\theta)\prod_{t=2}^{T}p(x_t|x_{t-1},\theta)\prod_{t=1}^{T}p(y_t|x_t,\theta)\right] \tag{62}$$

$$= \underbrace{\int \mathrm{d}x_1\, p(x_1|y,\theta)\frac{\mathrm{d}}{\mathrm{d}\theta}\log p(x_1|\theta)}_{=L_1}$$

$$+ \sum_{t=2}^{T} \underbrace{\int \mathrm{d}x_{t,t-1} p(x_t, x_{t-1}|y, \theta) \frac{\mathrm{d}}{\mathrm{d}\theta} \log p(x_t|x_{t-1}, \theta)}_{=L_2}$$

$$+ \sum_{t=1}^{T} \underbrace{\int \mathrm{d}x_t p(x_t|y, \theta) \frac{\mathrm{d}}{\mathrm{d}\theta} \log p(y_t|x_t, \theta)}_{=L_3} \tag{63}$$

We will find the explicit form for $L_1$, $L_2$ and $L_3$ below.

- Since $p(x_1|\theta) = \mathcal{N}(x_1; \mathbf{0}, K_0)$ and $p(x_1|y, \theta) = N(x_1; \mu_1, \Sigma_1)$ we have,

$$\log p(x_1|\theta) = -\frac{D_1 \log(2\pi)}{2} - \frac{1}{2} \log |K_0| - \frac{1}{2} x_1^{\mathsf{T}} K_0^{-1} x_1, \tag{64}$$

  hence,

$$\frac{\mathrm{d}}{\mathrm{d}\theta} \log p(x_1|\theta) = -\frac{1}{2} \frac{\mathrm{d}}{\mathrm{d}\theta} \log |K_0| - \frac{1}{2} x_1^{\mathsf{T}} \frac{\mathrm{d}K_0^{-1}}{\mathrm{d}\theta} x_1, \tag{65}$$

  and therefore

$$L_1 = -\frac{1}{2} \frac{\mathrm{d}}{\mathrm{d}\theta} \log |K_0| - \frac{1}{2} \operatorname{Tr}(\frac{\mathrm{d}K_0^{-1}}{\mathrm{d}\theta} \Sigma_1) - \frac{1}{2} \mu_1^{\mathsf{T}} \frac{\mathrm{d}K_0^{-1}}{\mathrm{d}\theta} \mu_1. \tag{66}$$

- Since $p(x_t|x_{t-1}, \theta) = \mathcal{N}(x_t; A_t x_{t-1}, Q_t)$ and $p(x_t, x_{t-1}|y, \theta) = N\left(\left[\begin{smallmatrix} x_t \\ x_{t-1} \end{smallmatrix}\right]; \left[\begin{smallmatrix} \mu_2 \\ \mu_1 \end{smallmatrix}\right], \left[\begin{smallmatrix} \Sigma_{22} & \Sigma_{21} \\ \Sigma_{12} & \Sigma_{11} \end{smallmatrix}\right]\right)$ we have,

$$\frac{\mathrm{d}}{\mathrm{d}\theta} \log p(x_t|x_{t-1}, \theta) = \frac{\mathrm{d}}{\mathrm{d}\theta} \left[ -\frac{1}{2} \log |Q_t| - \frac{1}{2} x_t^{\mathsf{T}} Q_t^{-1} x_t + x_t^{\mathsf{T}} Q_t^{-1} A_t x_{t-1} - \frac{1}{2} x_{t-1}^{\mathsf{T}} A_t^{\mathsf{T}} Q_t^{-1} A_t x_{t-1} \right]. \tag{67}$$

  Therefore $L_2 = L_{21} + L_{22} + L_{23} + L_{24}$, where

$$L_{21} = \int \mathrm{d}x_{t,t-1} p(x_t, x_{t-1}|y, \theta) \frac{\mathrm{d}}{\mathrm{d}\theta} \left( -\frac{1}{2} \log |Q_t| \right) \tag{68}$$

$$= -\frac{1}{2} \frac{\mathrm{d}}{\mathrm{d}\theta} \log |Q_t|, \tag{69}$$

$$L_{22} = \int \mathrm{d}x_{t,t-1} p(x_t, x_{t-1}|y, \theta) \frac{\mathrm{d}}{\mathrm{d}\theta} \left( \frac{-1}{2} x_t^{\mathsf{T}} Q_t^{-1} x_t \right) \tag{70}$$

$$= -\frac{1}{2} \operatorname{Tr}(\frac{\mathrm{d}Q_t^{-1}}{\mathrm{d}\theta} \Sigma_{22}) - \frac{1}{2} \mu_2^{\mathsf{T}} \frac{\mathrm{d}Q_t^{-1}}{\mathrm{d}\theta} \mu_2, \tag{71}$$

$$L_{23} = \int \mathrm{d}x_{t,t-1} p(x_t, x_{t-1}|y, \theta) \frac{\mathrm{d}}{\mathrm{d}\theta} \left( \frac{-1}{2} x_{t-1}^{\mathsf{T}} A_t^{\mathsf{T}} Q_t^{-1} A_t x_{t-1} \right) \tag{72}$$

$$= -\frac{1}{2} \operatorname{Tr}(\frac{\mathrm{d}}{\mathrm{d}\theta} (A_t^{\mathsf{T}} Q_t^{-1} A_t) \Sigma_{11}) - \frac{1}{2} \mu_1^{\mathsf{T}} \frac{\mathrm{d}}{\mathrm{d}\theta} (A_t^{\mathsf{T}} Q_t^{-1} A_t) \mu_1, \tag{73}$$

$$L_{24} = \int \mathrm{d}x_{t,t-1} p(x_t, x_{t-1}|y, \theta) \frac{\mathrm{d}}{\mathrm{d}\theta} \left( x_t^{\mathsf{T}} Q_t^{-1} A_t x_{t-1} \right) \tag{74}$$

$$= \int \mathrm{d}x_{t-1} p(x_{t-1}|y, \theta) \left[ \int \mathrm{d}x_t p(x_t|x_{t-1}, y, \theta) \left( x_t^{\mathsf{T}} \frac{\mathrm{d}}{\mathrm{d}\theta} (Q_t^{-1} A_t) x_{t-1} \right) \right] \tag{75}$$

$$= \int \mathrm{d}x_{t-1} p(x_{t-1}|y, \theta) \left[ [\mu_2 + \Sigma_{21} \Sigma_{11}^{-1} (x_1 - \mu_1)]^{\mathsf{T}} \frac{\mathrm{d}}{\mathrm{d}\theta} (Q_t^{-1} A_t) x_{t-1} \right] \tag{76}$$

$$= [\mu_2 - \Sigma_{21} \Sigma_{11}^{-1} \mu_1]^{\mathsf{T}} \frac{\mathrm{d}}{\mathrm{d}\theta} (Q_t^{-1} A_t) \mu_1$$

$$+ \operatorname{Tr} \left[ \Sigma_{11}^{-1} \Sigma_{12} \frac{\mathrm{d}}{\mathrm{d}\theta} (Q_t^{-1} A_t) \Sigma_{11} \right] + \mu_1^{\mathsf{T}} \Sigma_{11}^{-1} \Sigma_{12} \frac{\mathrm{d}}{\mathrm{d}\theta} (Q_t^{-1} A_t) \mu_1. \tag{77}$$

- Since $p(y_t|x_t, \theta) = \mathcal{N}(y_t; C_t x_t, R_t)$ and $p(x_t|y, \theta) = N(x_t; \mu_1, \Sigma_1)$, similarly as above, we obtain,

$$
\begin{aligned}
L_3 = -\,&\frac{1}{2}\frac{\mathrm{d}}{\mathrm{d}\theta}\log|R_t| - \frac{1}{2}y_t^\mathsf{T}\frac{\mathrm{d}R_t^{-1}}{\mathrm{d}\theta}y_t + y_t^\mathsf{T}\frac{\mathrm{d}}{\mathrm{d}\theta}(R_t^{-1}C_t)\mu_1 \\
&- \frac{1}{2}\operatorname{Tr}\left(\frac{\mathrm{d}}{\mathrm{d}\theta}(C_t^\mathsf{T}R_t^{-1}C_t)\Sigma_1\right) - \frac{1}{2}\mu_1^\mathsf{T}\frac{\mathrm{d}}{\mathrm{d}\theta}(C_t^\mathsf{T}R_t^{-1}C_t)\mu_1.
\end{aligned}
\tag{78}
$$

# 5  Experimental results

## 5.1  Synthetic data regression experiment

We generated a synthetic dataset that has 40k datapoints (2 dimensional inputs) using a GP with relatively short lengthscales and Gaussian noise. Blocks of datapoints are chosen at random to be our regression targets. We used an exponentiated quadratic kernel with ARD lengthscales. Our method achieved a better predictive performance, as measured by the standardized mean squared error (SMSE) and mean standardized log loss (MSLL), in a smaller training time and test time as shown in figures 3 and 4. For the tree-structured and local approximations, the data point labels are the number of clusters and the total number of inducing inputs respectively. For other approximations, the labels are the number of inducing inputs used for the experiment. We do not include the result for the DTC approximation to keep the graph clear. The result for the SSGP approximation is outside of the result regions being shown.

Figure 3: Synthetic dataset regression result: SMSE as a function of training time and test time per datapoint for different approximations.

Figure 4: Synthetic dataset regression result: MSLL as a function of training time and test time per datapoint for different approximations.

## 5.2 Audio data experiment

We performed imputatation tasks using a filtered signal and a real noisy signal. Typical results are shown in figures 5, 7 and **??**. Figure **??** shows the linearity of inference time as a function of dataset size using the tree-structured inducing-input approximation.

Figure 5: Filling missing data results using one component

Figure 6: Filling missing data results: we used five components in the spectral mixture and performed denoising and filling missing data for a filtered audio signal.

Figure 7: Filling missing data results: we used nine components in the spectral mixture and a real noisy audio signal with missing samples.

Figure 8: Spectral mixture experimental result: Inference time as a function of dataset size for different numbers of spectral mixture components