[Reviews · NeurIPS 2014]

Submitted by Assigned_Reviewer_12

1. Summary
This paper proposes a sparse method to (approximate) large scale inference in Gaussian process (GP) regression models. The main idea builds upon the inducing-point formalism underpinning most sparse methods for GP inference. As the computational cost of traditional sparse methods in GPs based on inducing points is O(NM^2), where N is the number of observations and M is the number of inducing points, the paper addresses the problem of large-scale inference by making conditional independence assumptions across inducing points. More specifically, the method proposed in the paper can be seen as a modified version of the partially independent conditional (PIC) approach, where not only the latent functions are grouped in blocks but also the inducing points are clustered in blocks (corresponding to those latent functions) and statistical dependences across inducing point blocks are modeled with a tree.
These additional independence assumptions make the resulting inference algorithm much more scalable as it only scales (potentially) linearly with the number of observations and the number of inducing points.

The method is evaluated on 1D and 2D problems showing that it outperforms standard sparse GP approximations.

2. Quality

The paper is technically sound in that it builds upon well-know technically sound approaches such as GP sparse approximations based on inducing points. The claim of linear computational cost as a function of the number of observations and/or the number of inducing points is correct when having very small block-sizes, as the cost is cubic on the size of the block.

However, it is surprising that the authors do not discuss what is potentially lost by the conditional independence assumptions underlying their approximation. The whole idea of GPs is to be able to model highly coupled observations, i.e. long-term dependencies. There is a lot to loose by assuming a “block-tree” structure in the inducing points, which is equivalent to making sparsity assumptions in their covariance. Their 1D and 2D experiments are not entirely convincing of the effectiveness of their approach to more general problems (see item 5 below).

There are also fundamental parts in the inference algorithm that are not obtained from first principles in a single probabilistic framework. For example:
(a) The partitioning of the data (and the inducing inputs) into blocks
(b) The selection/learning of the inducing points
(c) The learning of the tree structure using distances (although the algorithm for learning is sound)
All these are performed in an ad-hoc manner, which is uncharacteristic of a NIPS paper.

A fundamental criticism to this paper is that it appears to refer to traditional sparse GPs such as FITC, etc, as if they were state-of-the-art in sparse GP approximations. Clearly this is not the case. For example,
(i) The learning of inducing points have done before in sparse GP models and has been shown to be advantageous [Snelson and Ghahramani, 2006; Titsias, 2009].
(ii) The work in [Hensman et al, 2013] has shown that scalability at very large datasets is possible and that these should be the new benchmarks for sparse GPs.

In terms of completeness, although the technique developed is not specific to 1D or 2D, the experiments are only carried out on these types of applications. Hence, the paper will greatly benefit from experiments on higher dimensional spaces.

3. Clarity
Overall, the paper is well written and organized. The introduction of sparse GP approximations as a three-step process and the use of the KL divergence in the 3rd step is at best uninformative.
For example, there is nothing mysterious (line 110) to sparse GPs understood within a common probabilistic framework [Quinonero-Candela et al, 2005]. The introduction of the KL divergence is unnecessary and distracting of the main objective of the paper. Approximations such as FITC, PITC, PIC make some assumptions in the prior and of course these can also be written as a KL between the approximate prior and the true prior. Variational methods (such as VFE) and EP are, by definition, optimizing a KL divergence but such methods are more concerned with dealing with analytically intractable posteriors.

In terms of reproducibility, the paper does not provide enough detail for an expert reader to reproduce the experimental results. There is not detail regarding how the benchmark algorithms were executed (optimization details, etc) and vague sentences are used when referring to the execution of the proposed algorithm (e.g. “until a convergence criteria or a maximum number of function evaluations was reached”).

4. Originality
Neither the problem or the approaches presented in the paper is new. Standard regression problems with Gaussian noise are addressed, and the method proposed builds upon well-know sparse approximation techniques for GPs. The main computational advantage comes from making additional conditional independence assumptions across inducing points, whose impact is not discussed in the paper. As mentioned in item 2 (Quality) above, the authors seem unaware of the state of the art regarding GP approximations.

5. Significance
The results presented in the paper are very limited as they only concern 1D and 2D regression problems. the paper will greatly benefit from experiments on higher dimensional spaces. It is not clear how the paper will advance the state of the art as it does not compare with the most recent advances in GP approximations.

6. References
[Hensman et al, 2013]. Gaussian processes for big data. In UAI, 2013.
[Quinonero-Candela et al, 2005] A Unifying View of Sparse Approximate Gaussian Process Regression. in JMLR 2005.
[Snelson and Ghahramani, 2006] Sparse Gaussian Processes using Pseudo-inputs. In NIPS 2006.
[Titsias, 2009] Variational Learning of inducing variables in Sparse Gaussian Processes. In AISTATS 2009.
Summary: This paper proposes a method for sparse GP approximations based on well-known techniques such as the inducing points but comes short when comparing to recent advances in the area.

Submitted by Assigned_Reviewer_24

The authors propose a novel approximate inference scheme for Gaussian processes. The idea is to introduce pseudo data points that have tree-structured dependencies, making inference cheap.
The authors propose various heuristics on how to divide the data into smaller groups, either regular binning on a grid, or k-means clustering on input dimensions. Then Kruskal's algorithm is used to build the tree. Instead of optimizing over the pseudo inputs, the authors rather introduce a larger number of pseudo inputs (as message passing on a tree only scales linearly, not cubic as in other sparse approximations).

The quality of the paper is high and the method development is clear, with a nicely organized supplement. The method is relevant, especially in very low-dimensional settings, where local dependencies are quite clear such as time-series and spatial regression (as considered in the experimental section). Experimental evaluations are well executed.

further comments:

Please write explicitly how you group unlabeled data during prediction.
Even though there are some clear choices, it would be helpful to write it down explicitly.

>>A8: The test points are assigned to the closest training data block using Euclidean distance.
Thanks, please add this to the paper.

There is a lot of work on tree-based approximations to related Gaussian models that seem worth discussing, e.g. the paper by Wainwright, Sudderth, and Willsky, Nips 2000.

It would be good to get more details on how the comparison methods have been implemented an how they have been tuned. Sometimes the performance of these look unrealistically weak.

Summary: The paper is clear, has a very nice flow and bears sufficient novelty. The method should be of relevance for time-series and spatial applications and performs well in experiments.

Submitted by Assigned_Reviewer_32

This paper suggests a new technique for constructing low rank Gaussian process approximations based on pseudo-datasets. The approximation allows tree structured dependence among the pseudo-datapoints and this enables large numbers of pseudo-datapoints to be handled. This is important for some time series and spatial datasets in particular.

I think this is a worthy and original contribution to the literature; the paper is on the whole clearly written and it addresses an important problem.

Some minor technical questions and comments are given below.

1. In the experimental evaluations, different methods are being compared for the different experiments - I'm sure there are some good reasons but I'd like to understand why. In particular I would have thought that in experiment 2 the sparse spectral approximation would be a natural approach for this spectral mixture problem, even though it performs poorly in experiment 1 and is apparently not considered further for that reason.
2. The authors comment that their approach doesn't require stationarity, but to model non-stationarity within their model a parametric form for the non-stationarity would be required, and a suitable form might be difficult to specify. I think for problems where non-stationarity is a feature some of the local methods might perform relatively better if the learning of hyperparameters is also suitably localized, boundary effects not withstanding.
3. In Section 2.1 the authors describe their algorithm for selecting the pseudo-inputs and constructing the tree. After constructing an acyclic graph they suggest choosing one node to be the root at random. It is stated that the choice of the root does not affect the inference - I wasn't clear whether this statement was based on a theoretical argument or empirical experience and I wonder if this could be clarified.
Summary: I think this is a worthy and original contribution to the literature on Gaussian process approximation methods; the paper is on the whole clearly written, it addresses an important problem and unifies and extends some existing methods in a useful way.
Author Feedback
Author rebuttal: We thank the reviewers for their comments. Before responding to each review, we address a common concern that the experimental comparisons were light on detail. We closely followed the GP evaluation framework laid out in ref. [17], but acknowledge that we should have provided more detail, despite space constraints. Code at submission time was taken from the following sources: FITC from the GPML package; SSGP from Lázaro-Gredilla’s webpage; VFE from Titsias’s webpage; local, tree and chain methods are our own implementation. After submission we noticed that the VFE test time was larger than expected. Reimplementing this part of the code led to a revised time close to FITC. We will release the code, fix this minor issue in the paper and add further details, if accepted.

Reviewer_12
Q1: low dimensional datasets
A1: It would be very interesting to perform experiments on datasets with high dimensional input spaces. However, we expect that the new approximation is most useful for low dimensional datasets like time-series and spatial data. Whilst this might seem limiting from the general perspective of GP regression, these subfields are large and important parts of statistics that regularly employ GPs. The experiments were targeted in this direction and, in our opinion, convincingly demonstrate that the approximation scheme is effective in these domains, suggesting that it will have large impact.

Q2: learning inducing inputs
A2: Optimising pseudo-inputs is an important technique that can improve performance, especially when data form clusters in the input space that can be locally summarised. In the application domains considered in our paper, such as time-series, the data are often unclustered or weakly clustered. Optimisation therefore leads to only modest improvement in accuracy over methods such as regular spacing or sub-sampling. Instead, it is key to use very large numbers of pseudo points (e.g. for a minute of audio the argument in section 1.3 suggests that 10^4-10^5 pseudo points will be required regardless of whether input optimisation is used). Apart from the local method, no existing GP approximation meets this requirement. More generally, allocating compute time between pseudo-input optimisation and hyper-parameter optimisation is an important unsolved problem [17] that makes it hard to perform the all important speed-accuracy trade-off comparisons.

Q3: ad-hoc data partitioning and tree construction
A3: We agree that the formation of the tree is the least principled part of the paper, deserving attention in future work. However, it is common practice to use heuristic schemes of this ilk (e.g. the clustering schemes employed in [17] & [5]) and we find in practice that they work well.

Q4: when does the approximation work well and what is lost
The approximation is well suited to datasets with, (a) long range correlations, (b) large input domains (many 'length-scales' in extent), (c) low dimensional input spaces. (a) means that large number of datapoints can be summarised in a block containing a small number of pseudo-points. (b) means that the data can be split into many blocks. (c) means that trees can be built which preserve local correlations. In such cases the blocks can be chosen to span several length-scales and experiments show that the method performs essentially identically to the full GP. Note that conditional independencies implied by the tree result in a sparse precision matrix (not covariance as the reviewer states). Widely separated blocks can have strong correlations meaning the method degrades gracefully.

Q6: unnecessary introduction of the KL divergence
A6: As the reviewer correctly points out ref. [3] shows a number of approximation schemes can be obtained by specifying a new generative model produced by removing dependencies in the GP. However, once the dependencies are removed, it does not provide a constructive procedure for choosing the parameters of that simplified model. Sollich and Snelson later realised that the choice made in FITC/PITC could be justified as a KL minimisation (see Snelson’s thesis, pg 54). This calibration step is critical for the development of the tree-structured model as it provides a way of deriving the tree parameters. In principle, alternative calibration steps could be considered e.g. KL[p(f)||q(f)]. The KL arguments are therefore central.

Q7: the lack of comparisons to/unaware of state-of-the-art
A7: We are aware of the state-of-the-art. The experiments in the paper compared to four methods including those of Snelson and Ghahramani, 2006 (FITC) and Titsias, 2009 (VFE). We have discussed in A2 the reasons why we do not consider the pseudo-input optimisation variants. Hensman et al, 2013 is a partial E-step version of VFE and so we expect the VFE to be more accurate, but with a larger time-complexity as the new method scales as O(M^3). This means that whilst it is well suited to large datasets that can be summarised using small M, it will perform poorly in the applications we consider where M must be very large.

Reviewer_24
Q8: assignment of unlabeled data during prediction
A8: The test points are assigned to the closest training data block using Euclidean distance.

Q9: Sudderth et al. 2000 paper
A9: Thanks, the paper is very relevant.

Q10: performance of comparison methods looks unrealistically weak
A10: The current methods do not work well for the applications we are considering. For example, their performance can be made arbitrarily poor by selecting a long enough time-series.

Reviewer_32
Q11: different methods are being compared for the different experiments
A11: At submission time we did not have non-standard kernels implemented for VFE and SSGP. We do now and will include them in the final version. Results mirror the SE cases.

Q13: choice of the root node does not affect the inference
A13: This is a theoretical result, see Koller and Friedman 2009, pg 354.